# Assessing the relationship between agency and peer violence among adolescents aged 10 to 14 years in Kinshasa, Democratic Republic of Congo and Blantyre, Malawi: A cross-sectional study

**Astha Ramaiya**[1]*, **Linnea Zimmerman**[1], **Eric Mafuta**[2], **Aimee Lulebo**[2], **Effie Chipeta**[3], **William Stones**[3], **Caroline Moreau**[1,4]

**1** Department of Population, Family and Reproductive Health, Bloomberg School of Public Health, Johns Hopkins University, Baltimore, Maryland, United States of America, **2** Faculty of Medicine, Kinshasa School of Public Health, University of Kinshasa, Kinshasa, Democratic Republic of Congo, **3** University of Malawi College of Medicine, Blantyre, Malawi, **4** Soins Primaires et Prévention, Center for Research in Epidemiology and Population Health (CESP), Villejuif, France

* aramaiy1@jhu.edu

## Abstract

### Background

Interpersonal violence has physical, emotional, educational, social, and economic implications. Although there is interest in empowering young people to challenge harmful norms, there is scant research on how individual agency, and, specifically, the "power to" resist or bring about an outcome relates to peer violence perpetration and victimization in early adolescence. This manuscript explores the relationship between individual agency and peer violence perpetration and victimization among very young adolescents (VYAs) living in two urban poor settings in sub-Saharan Africa (Kinshasa, Democratic Republic of Congo (DRC) and Blantyre, Malawi).

### Methods and findings

The study draws on two cross-sectional surveys including 2,540 adolescents 10 to 14 years from Kinshasa in 2017 (girls = 49.8% and boys = 50.2%) and 1,213 from Blantyre in 2020 (girls = 50.7% and boys = 49.3%). The sample was school based in Malawi but included in-school and out-of-school participants in Kinshasa due to higher levels of early school dropout. Peer violence in the last 6 months (dependent variable) was defined as a four categorical variable: (1) no victimization or perpetration; (2) victimization only; (3) perpetration only; and (4) both victimization and perpetration. Agency was operationalized using 3 scales: freedom of movement, voice, and decision-making, which were further divided into tertiles. Univariate analysis and multivariable multinomial logistic regressions were conducted to evaluate the relationships between each agency indicator and peer violence. The multivariable regression adjusted for individual, family, peer, and community level covariates. All analyses were stratified by gender and site.

**Data Availability Statement:** The data used for the analyses and codebook are available as S1 Data and S1 Codebook, respectively.

**Funding:** This study was conducted with support from the UNDP-UNFPA-UNICEF-WHO-World Bank Special Programme of Research, Development and Research Training in Human Reproduction (HRP), a co-sponsored programme executed by the World Health Organization (WHO). Support for each Global Early Adolescent Study site is provided by the Bill & Melinda Gates Foundation [OPP1197258] and the United States Agency for International Development [AID-OAA-A-15-00042] in Kinshasa and Rutgers, The Netherlands for Blantyre, Malawi. The funders had no role in study design, data collection and analysis, decision to publish, or preparation of the manuscript.

**Competing interests:** The authors have declared that no competing interests exist.

**Abbreviations:** ACEs, adverse childhood experience; COVID-19, Coronavirus Disease 2019; DRC, Democratic Republic of Congo; GEAS, Global Early Adolescent Study; RRR, relative risk ratio; SD, standard deviation; STROBE, Strengthening the Reporting of Observational Studies in Epidemiology; VYA, very young adolescent.

In both sites, adolescents had greater voice and decision-making power than freedom of movement, and boys had greater freedom of movement than girls. Boys in both settings were more likely to report peer violence in the last six months than girls (40% to 50% versus 32% to 40%, $p < 0.001$), mostly due to higher rates of a perpetration–victimization overlap (18% to 23% versus 10% to 15%, $p < 0.001$). Adolescents reporting the greatest freedom of movement (Tertile 3) (with the exception of girls in Kinshasa) had a greater relative risk ratio (RRR) of reporting a perpetrator–victim overlap (boys Kinshasa: RRR = 1.9 (1.2 to 2.8, $p = 0.003$); boys Blantyre: RRR = 3.8 (1.7 to 8.3, $p = 0.001$); and girls Blantyre: RRR = 2.4 (1.1 to 5.1, $p = 0.03$)). Adolescents with the highest decision-making power in Kinshasa also had greater RRR of reporting a perpetrator–victim overlap (boys: RRR = 3.0 (1.8 to 4.8, $p < 0.001$). Additionally, girls and boys in Kinshasa with intermediate decision-making power (tertile 2 versus 1) had a lower RRR of being victimized (Girls: RRR = 1.7 (1.02 to 2.7, $p = 0.04$); Boys: RRR = 0.6 (0.4 to 0.9, $p = 0.01$)). Higher voice among boys in Kinshasa (Tertile 2: RRR = 1.9 (1.2 to 2.9, $p = 0.003$) and Tertile 3: 1.8 (1.2 to 2.8, $p = 0.009$)) and girls in Blantyre (Tertile 2: 2.0 (1.01 to 3.9, $p = 0.048$)) was associated with a perpetrator–victim overlap, and girls with more voice in Blantyre had a greater RRR of being victimized (Tertile 2: RRR = 1.9 (1.1 to 3.1, $p = 0.02$)). Generally, associations were stronger for boys than girls, and associations often differed when victimization and perpetration occurred in isolation of each other. A main limitation of this study is that the cross-sectional nature of the data does not allow a causal interpretation of the findings, which need further longitudinal exploration to establish temporality.

## Conclusions

In this study, we observed that peer violence is a gendered experience that is related to young people's agency. This stresses the importance of addressing interpersonal violence in empowerment programs and of including boys who experience the greatest perpetration–victimization overlap.

Author summary

### Why was this study done?

- The Convention on the Rights of the Child prevents all forms of violence against children.

- Bullying ranges anywhere from 10% to 65% among adolescents between 11 and 15 years.

- In order to reduce violence, programs focus on empowering girls by increasing agency.

- There is a lack of studies that assess the relationship between individual agency and peer violence in early adolescence and examine this association by gender and across different social contexts. In addition, few studies examine the overlap between peer violence victimization and perpetration.

## What did the researchers do and find?

- We used data from an ongoing multicountry study on gender socialization and adolescent health to examine the relationship between agency and peer violence among adolescent boys and girls 10 to 14 years. Data were collected among in-school and out-of-school adolescents in Kinshasa, Democratic Republic of Congo (DRC) in 2017 and among in-school adolescents in Blantyre, Malawi in 2020.

- Agency consisted of 3 scales (freedom of movement, voice, and decision-making), each categorized into tertiles. Peer violence in the last six months was a four categorical variable distinguishing no violence, victimization only, perpetration only, and a victimization–perpetration overlap.

- Multivariate multinomial logistic regressions, stratified by site and gender, were conducted to estimate the relative risk ratio (RRR) of different forms of peer violence according to the three agency indicators adjusting for individual, interpersonal, and community factors.

- Results show that, generally, higher agency (greater voice, decision-making, or freedom of voice) is associated with an increased relative risk of reporting a perpetrator–victim overlap relative to no violence, especially among boys.

## What do these findings mean?

- Contrary to beliefs that boys are perpetrators and girls are victims, our study shows that gender differences mostly relate to an overlap of perpetration and victimization, more commonly reported by boys than girls. These results call for violence prevention strategies to break the cycle of violence contributing to the overlap of perpetration and victimization, especially for boys.

- Programs should focus on both genders when including empowerment components into interventions to reduce violence.

- The greater risk of an overlap of violence perpetration and victimization among adolescents with greater agency calls for the integration of violence prevention strategies into empowerment programs to prevent potential unexpected consequences.

- The greater RRR of violence perpetration and victimization among adolescents who have experienced adverse childhood experiences (ACEs) as well as the extent of a perpetration–victimization overlap reaffirms the importance of a trauma-informed care to prevent teen violence.

## Introduction

Interpersonal violence, the intentional use of force or power on a family member or unrelated individuals [1], is particularly salient in adolescence due to the significance of violent experiences on developmental trajectories, with considerable physical, emotional, educational, social, and economic implications [2–4]. Bullying is particularly prevalent in this stage of life, rising

in early adolescence, peaking in late adolescence, and decreasing in young adulthood [2,5,6]. Data from 144 countries participating in the Global School-based Health Survey show that 10% to 65% of adolescents 11 to 15 years report having been bullied in the past year [4]. These global figures mask significant variation by geography and gender [7], with high prevalence noted in the African region (43.5%, 43.0% to 44.3%) [8] and greater violence exposure among boys compared to girls [8,9].

Interpersonal violence, including peer violence perpetration and/or victimization, is informed by the interplay of personal and contextual factors [2,6]. At the individual level, age, gender, history of adverse childhood experiences (ACEs), medical/physical conditions that affect the nervous system, and psychological characteristic, i.e., impulsivity, attention deficit, and substance abuse are associated with increased risk of violence, including bullying and general aggression. At the family and peer levels, experiencing and/or witnessing harsh parenting, domestic violence, and antisocial peer behaviors increase the risk of adolescent violent behaviors. At the community level, community violence, lack of social cohesion, and cultural norms inform peer violence [6]. These socioecological factors intersect with structural factors, including poverty, racial/ethnic discrimination, and gender discrimination to inform violent behaviors and outcomes [10,11]. While these factors are traditionally examined in relation to perpetration and victimization separately, there is considerable similarity in the personal and social factors associated with perpetration and victimization, leading to a growing interest into the perpetrator–victim overlap [12].

Preventing all forms of violence is a fundamental human right enshrined in the Convention on the Rights of the Child and in the Sustainable Development Goal 16.2 [13]. Interventions to prevent violence generally address risks and protective factors in specific populations, such as adolescents, and/or structural determinants of inequities and discrimination [14]. At the intersection of these approaches, women and girl's empowerment is a particular mechanism invoked to reduce violence against women, by increasing women's agency, while challenging hegemonic forms of masculinity that promote male dominance and aggression through greater bargaining power [15,16]. A number of studies support raising girls' voice and agency to prevent gender-based violence [17], but a few studies also alert to the potential backlash associated with challenging the gender order by raising women's autonomy and reducing male dominance, which has, in some cases, resulted in increased gender-based violence [18–20].

While there is growing interest in empowering young people to challenge harmful norms and become the agents of their own lives, there is little understanding of how agency relates to peer violence perpetration and victimization in this stage of the life course, when gendered expectations intensify, encouraging young girls to stay home and to rely on others to meet their needs, while promoting boys' independence and emancipation [21]. Agency is defined as the ability to set and achieve one's goals by mobilizing resources [22,23]. Agency is multidimensional including "the power to" or the ability to act according to one's goals, "the power over" or the ability to influence others, and the "power with" or the ability to exercise collective action [24]. It touches on all spheres of life and expands significantly in this critical stage of the life course [25–27]. While current efforts focus on collective empowerment to prevent peer violence, less is known about the potential of individual agency ("power to") to refrain from or engage in peer violence in early adolescents and how these associations differ by gender and by social context.

The Global Early Adolescent Study (GEAS), a multisite investigation of gender socialization and adolescent health and well-being, offers a unique opportunity to explore these associations, using validated multidimensional measures of agency exploring freedom of movement, voice, and decision-making power among very young adolescents (VYAs). We used these measures to assess the relationship between agency and peer violence perpetration and

victimization among poor urban VYA in Kinshasa, Democratic Republic of Congo (DRC) and Blantyre, Malawi. We hypothesize that boys and girls with more freedom of movement would have greater exposure to violence (victimization and perpetration) and that greater voice and decision-making among boys would be associated with violence perpetration. We anticipate a more complex relationships for girls as greater agency could increase their ability to resist aggression but could also increase victimization if they do not conform to the gender order, as suggested in the systematic review of empowerment programs for adolescent girls [17].

## Methods

This study is reported as per the Strengthening the Reporting of Observational Studies in Epidemiology (STROBE) guideline (S1 STROBE Checklist).

### Sampling and recruitment

The GEAS is a prospective study currently conducted in 10 countries (12 sites) across five continents. In this study, we used baseline (cross-sectional) data from the 2 GEAS sites in sub-Saharan Africa: Kinshasa, DRC and Blantyre, Malawi. S1 IRB and S2 IRB outline the Institutional Review Board (IRB) submission and study protocol for both sites. Both sites used the same survey instruments and eligibility criteria (adolescents aged 10 to 14 years who provided adolescent assent and parental consent to participate and who were able to understand the questionnaire), but sampling and data collection procedures differed.

In Kinshasa, in-school and out-of-school adolescents were selected form two urban poor communes selected by Save the Children for an intervention (Growing Up Great!). Schools were purposively selected in the intervention and control zones after stratification by commune and school type (public school, religiously affiliated non-for-profit schools, and private religious schools). In the intervention arm, 25 students per school were selected from the Growing Up Great! school clubs. In the control arm, 25 adolescents were randomly selected after stratification by gender and age. Out-of-school adolescents were recruited from the same communes and randomly selected from a listing of households. A total of 2,840 adolescents (2,016 in school and 826 out of school) were selected and provided assent and parental consent to participate. All participants who were approached completed the survey.

In Blantyre, sampling was a two-stage process with the selection of schools and students for an intervention (Very Young Adolescence 2.0), which was ultimately canceled due to the Coronavirus Disease 2019 (COVID-19) pandemic. Eight public urban schools that offered sixth grade were listed and purposively allocated to "intervention" or "control" status in consultation with the city education authorities and head teachers. Class registers for sixth grade in each of the schools were used to select between 200 and 550 adolescents per school depending on number of students at each school. The sample for the present analysis includes 1,694 participants, out of the 2,140 students who had been approached before the COVID-19 shutdown.

In both sites, cases missing more than 15% of data were dropped ($n$ = 10 (0.4%) in Kinshasa and $n$ = 139 (8.2%) in Malawi). In addition, 27 (0.9%) respondents in Kinshasa and 53 (3.1%) respondents in Blantyre were excluded due to missing information on the main independent and dependent variables. The analytical samples included 3,753 adolescents ($n$ = 2,540 in Kinshasa and $n$ = 1,213 in Blantyre).

Data collection occurred in schools and/or community centers in November 2017 in Kinshasa and February to March 2020 in Blantyre. In Kinshasa, adolescents were interviewed face to face by trained interviewers due to low literacy rates, while in Blantyre, adolescents self-completed the survey using Audio Computer-Assisted Self-Interview features. All surveys were uploaded to the SurveyCTO server and compiled into deidentified datasets. The study

received ethical approval from both sites (Kinshasa School of Public Health and College of Medicine, University of Malawi). The study was deemed exempt for secondary data collection by the Johns Hopkins School of Public Health's IRB.

## Measures

S1 Table outlines the questions and responses used from the GEAS questionnaire to answer the research question in this manuscript.

Our outcome measure of peer violence perpetration and victimization was based on responses to the following two questions: (i) During the last six months, have you slapped, hit, or otherwise physically hurt another boy or girl in a way that they did not want? (ii) During the last six months, have you been slapped, hit, or otherwise physically hurt by another boy or girl in a way that you did not want? We created a four-level categorical variable: "no peer violence," "victimization alone," "perpetration alone," and "perpetration and victimization." This classification was intended to identify the high level of overlap between perpetration and victimization while isolating differential influences of agency on each component. While the GEAS study collected information about the gender of the protagonist, we did not include this information in the indicator due to small sample size concerns. However, we report the distribution of violence by respondent and protagonist gender in the result section.

We used the GEAS cross-culture measures of agency that reflect three common dimensions of women's empowerment (freedom of movement, voice, and decision-making power), which are salient in early adolescence. The dimensions were adapted to the developmental stage of early adolescence, when young people have limited autonomy. The measures were validated in formative research across 11 settings in 5 continents [28]:

- freedom of movement (5 items) measuring young people's ability to circulate in their community without adult supervision (Polychoric Alpha of 0.71 in Kinshasa and 0.87 in Blantyre);

- voice (7 items) assessing young people's ability to express their needs and opinions (Polychoric Alpha of 0.78 in Kinshasa and 0.85 in Blantyre); and

- decision-making (4 items) evaluating young people's influence over daily decisions (Polychoric Alpha of 0.75 in Kinshasa and −0.77 in Blantyre).

The items were scored on a 4-point Likert scale and summarized into three continuous measures. The measures were not normally distributed, and, therefore, we created score tertiles, based on the distribution of the indicator by site. Correlations between the three agency scores ranged from 0.1 to 0.6.

Following Blum and Heise's ecological models [10,29], we considered individual, family, peer, school, and neighborhood covariates that were related to interpersonal violence [11]. At the individual level, we explored age, educational attainment (appropriate grade for age), perceptions of gender stereotypical traits, and history of ACEs (none, 1, 2, 3, and 4 or more). Gender stereotypical traits were captured in a 7-item scale assessing adolescent's perceptions of male toughness and female vulnerability [30]. Family factors included family structure (no parent, single parent, and both parent household), parent connectedness (assessed by the question "do you feel close to your main caregiver? By close, we mean you talk to that person and tell them about personal and important things," dichotomized as "a lot" or "less than a lot"), parental awareness/monitoring (assessed by caregiver's knowledge of adolescents' friends, school performance, and general whereabouts and dichotomized "yes" or "no"), and wealth assets (quintiles); peer factors considered peer composition (dichotomized as "same gender

friends" or "opposite gender friends") and time spent with friends (dichotomized as "every day" or "less than every day"). Finally, community level factors included neighborhood social cohesion (assessed by "how likely is it that an adult in your neighborhood would do something like intervene if children or teenagers were (a) damaging property; (b) spraying paint on walls (graffiti); (c) bullying or threatening another person; and (d) fighting with another person"). An additive index was created and dichotomized as "low" and "high" based on the median in each site.

## Analysis

We first described the distribution of the individual, family, peer, and community characteristics of the study population by gender and site and tested for gender differences using chi-squared tests or unpaired *t* tests (in the case of gender norm perceptions and agency scores). We then estimated the percentage of adolescents reporting violence perpetration and victimization in the last six months by gender and site and tested for gender differences using chi-squared tests. Next, we tested for differences in the distribution of the four categorical measure of peer violence by tertile of each agency measure, using chi-squared tests. Next, we ran multivariate multinomial logistic regression models to evaluate the relationships between agency indicators and violence victimization or perpetration or both, using no violence as the reference category and adjusting for all other covariates (individual, family, peer, and community factors) in each site. The models produce adjusted relative risk ratios (RRRs), which indicate how the risk of the outcome falling in one if the comparison groups (victimization, perpetration, or victimization/perpetration overlap) compared to the risk of being in the reference group (no violence) changes with one unit change of the independent variable (agency tertiles), after adjusting for other covariates [31]. All analyses were stratified by gender to investigate gender differences in the association between agency and peer violence experiences. Likewise, we stratified the analysis by site to account for cultural and study implementation differences. We tested for differences in the associations between agency and peer violence by school status in Kinshasa but found not overall interactions and therefore combined these groups into one. Finally, we pooled our data across sites and ran a multivariate multinomial logistic regression model among boys and girls separately to see if the associations between agency measures and violence differed by site. We tested the overall interactions for boys and girls using the command "testparm." All analysis were conducted using Stata 14.2 [32].

## Results

The characteristics of the study samples are presented in Table 1. A total of 2,540 adolescents were included in Kinshasa, evenly distributed between girls (49.8% (*n* = 1,266)) and boys (50.2% (*n* = 1,274)). A total of 1,213 adolescents responded in Blantyre with 50.7% of girls (*n* = 615) and 49.3% of boys (*n* = 598). A majority of adolescents, 64.1% in Kinshasa and 60.4% in Blantyre, were between 10 and 12 years old. In Kinshasa, 60% of adolescents were living with both parents, while this percentage dropped to 31% in Blantyre. Most adolescents had experienced ACEs, with 19% to 28% reporting 4 ACEs or more. Adolescents were more likely to fall below grade level in Blantyre, while 26% of girls and 31% of boys in Kinshasa were out of school. Adolescents generally perceived a high level of social cohesion in their community (70.5% to 74.3%). A majority of adolescents in both sites perceived unequal stereotypical gender traits (mean: 4.2 to 4.5 on a scale from 1 to 5), which considered boys as tough and girls as vulnerable.

Across sites, adolescents scored higher on scales of voice (range: 2.4 to 3.0 on a scale from 1 to 4) and decision-making (range: 2.7 to 3.0 on a scale from 1 to 4) than on freedom of

**Table 1. Sociodemographic, family, peer, and community characteristics by gender and by site.**

| | | Kinshasa, DRC | | | Blantyre, Malawi | | |
|---|---|---|---|---|---|---|---|
| | | **Girls** *n* = 1,266 | **Boys** *n* = 1,274 | *p*-value | **Girls** *n* = 615 | **Boys** *n* = 598 | *p*-value |
| **Age** | 10 to 12 years | 66.1% (837) | 62.0% (790) | 0.03 | 63.7% (392) | 57.0% (341) | 0.02 |
| | 13 to 14 years | 33.9% (429) | 38.0% (484) | | 36.3% (223) | 43.0% (257) | |
| **Educational attainment** | Out of school | 26.1% (330) | 30.9% (394) | | Data not available[a] | | |
| | Lower than age expected grade | 14.3% (181) | 14.2% (181) | 0.02 | 62.3% (383) | 73.8% (441) | <0.001 |
| | Age expected grade or higher | 59.6% (755) | 54.9% (699) | | 37.7% (232) | 26.3% (157) | |
| **ACEs** | 0 ACEs | 25.4% (322) | 18.7% (238) | | 31.9% (196) | 21.2% (127) | |
| | 1 ACEs | 21.7% (275) | 23.0% (293) | | 21.0% (129) | 21.6% (129) | |
| | 2 ACEs | 18.8% (238) | 21.0% (267) | 0.002 | 16.4% (101) | 16.1% (96) | <0.001 |
| | 3 ACEs | 12.5% (158) | 14.5% (185) | | 11.7% (72) | 12.7% (76) | |
| | 4 + ACEs | 21.6% (273) | 22.8% (291) | | 19.0% (117) | 28.4% (170) | |
| **Household structure** | Both parents | 59.2% (749) | 56.0% (714) | | 30.9% (190) | 38.5% (230) | |
| | One parent | 26.9% (340) | 29.5% (376) | 0.25 | 62.0% (381) | 53.5% (320) | 0.01 |
| | Grandparents and other | 14.0% (177) | 14.4% (184) | | 7.2% (44) | 8.0% (48) | |
| **High parental closeness[b]** | | 60.9% (771) | 62.2% (792) | 0.51 | 78.5% (483) | 74.4% (445) | 0.09 |
| **High parental monitoring and awareness about adolescent's friends, school performance, and whereabouts[b]** | | 50.2% (635) | 46.1% (587) | 0.04 | 98.4% (605) | 99.0% (592) | 0.34 |
| **Peer composition[b]** | Any opposite gender friends | 38.6% (489) | 45.8% (583) | <0.001 | 72.5% (446) | 80.3% (480) | 0.002 |
| **Time spent with friends[b]** | Nearly every day | 45.4% (575) | 59.0% (752) | <0.001 | Data not available[a] | | |
| **Social cohesion** | High | 70.5% (893) | 74.3% (946) | 0.04 | 71.5% (440) | 70.6% (422) | 0.71 |
| **Gender stereotypical trait[c]** | | 4.5 (0.6) | 4.4 (0.7) | <0.001 | 4.2 (0.8) | 4.3 (0.7) | <0.001 |
| **Agency (mean score on a scale 1 to 4, SD)** | Freedom of movement overall | 1.4 (0.6) | 1.8 (0.8) | <0.001 | 1.8 (0.7) | 2.3 (0.8) | <0.001 |
| | *Tertile 1* | 1.0 (0.01) | 1.0 (0.01) | 0.90 | 1.2 (0.2) | 1.2 (0.2) | 0.50 |
| | *Tertile 2* | 1.4 (0.2) | 1.5 (0.2) | <0.001 | 1.9 (0.2) | 1.9 (0.2) | 0.15 |
| | *Tertile 3* | 2.3 (0.5) | 2.5 (0.6) | <0.001 | 2.9 (0.5) | 3.1 (0.5) | 0.06 |
| | Voice overall | 2.4 (0.7) | 2.5 (0.7) | <0.001 | 2.9 (0.7) | 3.0 (0.7) | 0.009 |
| | *Tertile 1* | 1.7 (0.3) | 1.7 (0.3) | 0.09 | 2.1 (0.5) | 2.2 (0.5) | 0.21 |
| | *Tertile 2* | 2.5 (0.2) | 2.5 (0.2) | 0.61 | 3.1 (0.2) | 3.1 (0.2) | 0.12 |
| | *Tertile 3* | 3.2 (0.3) | 3.2 (0.3) | 0.37 | 3.7 (0.2) | 3.7 (0.2) | 0.10 |
| | Decision-making overall | 2.7 (0.9) | 2.7 (0.9) | 0.89 | 2.9 (0.8) | 3.0 (0.8) | 0.003 |
| | *Tertile 1* | 1.7 (0.4) | 1.7 (0.4) | 0.02 | 1.9 (0.5) | 1.9 (0.5) | 0.85 |
| | *Tertile 2* | 2.9 (0.3) | 2.9 (0.3) | 0.23 | 3.0 (0.2) | 3.0 (0.2) | 0.22 |
| | *Tertile 3* | 3.8 (0.2) | 3.8 (0.2) | 0.42 | 3.8 (0.2) | 3.8 (0.2) | 0.37 |
| **Perpetrated violence in the last six months** | No | 76.9% (973) | 71.0% (904) | <0.001 | 79.0% (486) | 69.7% (417) | 0.001 |
| | Yes, both boys and girls | 2.8% (35) | 6.7% (85) | | 5.7% (35) | 6.5% (39) | |
| | Yes, a boy or boys | 6.5% (81) | 17.4% (222) | | 6.0% (37) | 11.7% (70) | |
| | Yes, a girl or girls | 14.0% (177) | 5.0% (63) | | 9.3% (57) | 12.0% (72) | |
| **Victimized in the last six months** | No | 80.3% (1,016) | 70.8% (903) | <0.001 | 67.3% (414) | 57.2% (342) | |
| | Yes, both boys and girls | 1.4% (18) | 5.7% (73) | | 7.8% (48) | 6.9% (41) | <0.001 |
| | Yes, a boy or boys | 6.2% (79) | 18.8% (240) | | 12.5% (77) | 16.1% (96) | |
| | Yes, a girl or girls | 12.1% (153) | 4.6% (58) | | 12.4% (76) | 19.9% (119) | |
| **Peer violence composite** | No violence or victimization | 67.5% (854) | 60.1% (765) | <0.001 | 61.1% (376) | 49.5% (296) | <0.001 |
| | Victimization only | 9.4% (119) | 10.9% (139) | | 17.9% (110) | 20.2% (121) | |
| | Perpetration only | 12.8% (162) | 10.8% (138) | | 6.2% (38) | 7.7% (46) | |
| | Victimization and perpetration | 10.4% (131) | 18.2% (232) | | 14.8% (91) | 22.6% (135) | |

[a]Data not available because these questions were not included in the questionnaire.

[b]Parental closeness, parental monitoring and awareness, friend composition, and time spent with friends were collapsed and made into dichotomous variables based on the distribution of the data.

[c]Gender stereotypical traits score ranged from 1 to 5, with higher scores indicating more unequal perceptions of gender norms.

Chi-squared tests were used to estimate gender difference within each site for all variables except gender norm perceptions and agency, which used unpaired *t* test.

ACEs, adverse childhood experience; DRC, Democratic Republic of Congo; SD, standard deviation.

movement (range: 1.4 to 2.3 on a scale from 1 to 4). Overall, 18.2% of boys in Kinshasa and 22.6% of boys in Blantyre reported experiencing a perpetrator–victim overlap in the last six months, while this was true of 10.4% to 14.8% of girls in each site, respectively. Gender differences in perpetration or victimization alone were negligible (Kinshasa perpetration: $p = 0.12$, Kinshasa victimization: $p = 0.21$, Blantyre perpetration $p = 0.30$, and Blantyre victimization $p = 0.30$). Peer violence mostly involved same gender peers in Kinshasa, for example, among adolescents who were victimized, 64% of boys were victimized by boys only, and 61% of girls were victimized by girls only. Likewise, among adolescents who perpetrated violence, 60% of boys committed aggression against boys only, and 60% of girls committed aggression against girls only. Experiences were more diversified in Blantyre. Among adolescents who were victimized, 38% of boys were victimized by boys only and 46% by girls only, and 38% of girls were victimized by girls only and 38% were victimized by boys only. Among adolescents who perpetrated violence, 39% boys committed aggression on boys only, and 40% committed aggression against girls only, while 44% of girls committed aggression against girls only, and 29% perpetrated violence against boys only.

Table 2 shows the proportion of peer violence experience according to an adolescent's level of agency (in tertiles). Among boys in both sites and girls in Blantyre, there was a higher percentage of overlap of perpetration and victimization among those who expressed the greatest freedom of movement (24.0% among Kinshasa boys in Tertile 3 versus 12.5% in Tertile 1 $p < 0.001$; 33.2% among Blantyre boys in Tertile 3 versus 9.0% in Tertile 1 $p < 0.001$; and 26.7% among Blantyre girls in Tertile 3 versus 9.5% in Tertile 1 $p < 0.001$). Among boys in Kinshasa and girls in Blantyre, there were higher percentages of adolescents reporting a perpetrator–victim overlap among those with the most voice (22.2% among Kinshasa boys in Tertile 3 versus 11.2%, in Tertile 1 $p = 0.002$ and 16.4% among Blantyre girls in Tertile 3 versus 12.0% in Tertile 1 $p = 0.05$). Finally, among boys in Kinshasa and girls in both sites, there was a higher percentage of adolescents reporting a perpetrator–victim overlap among those with the most decision-making power (27.8% among Kinshasa boys in Tertile 3 versus 9.1% in Tertile 1 $p < 0.001$; 13.5% among Kinshasa girls in Tertile 3 versus 9.2% in Tertile 1 $p = 0.04$; and 16.6% among Blantyre girls in Tertile 3 versus 12.6% in Tertile 1 $p = 0.04$).

In the multivariate analyses (Table 3) including all confounding variables, boys with the greatest freedom of movement had elevated RRR of a perpetrator–victim overlap relative to those with the lowest freedom of movement in both sites (Kinshasa Tertile 3 versus Tertile 1: RRR = 1.9 (95% CI: 1.2 to 2.8), $p = 0.003$; Blantyre Tertile 2 versus Tertile 1: RRR = 2.4 (1.1 to 5.5), $p = 0.03$; and Blantyre Tertile 3 versus Tertile 1: RRR = 3.8 (1.7 to 8.3), $p = 0.001$). The same was true for girls with greater freedom of movement in Blantyre (Tertile 3 versus Tertile 1: RRR = 2.4 (1.1 to 5.1), $p = 0.03$). In addition, boys in Kinshasa with the greatest freedom of movement had an elevated relative risk of being victimized relative to those with the least freedom of movement (Tertile 3 versus Tertile 1: RRR = 2.8 (1.7 to 4.6), $p < 0.001$).

Girls in Blantyre with a greater ability to be heard (voice) had an elevated relative risk of being victimized (Tertile 2 versus Tertile 1: RRR = 1.9 (1.1 to 3.5), $p = 0.02$) and an elevated risk of reporting an overlap of perpetration and victimization in comparison to girls who had the least voice (Tertile 2 versus Tertile 1: RRR = 1.9 (1.01 to 3.9), $p = 0.048$). Boys in Kinshasa with a greater ability to be heard had an elevated relative risk of a violence perpetration and victimization overlap relative to those who with the least voice (Tertile 2 versus Tertile 1: RRR = 1.9 (1.2 to 2.9), $p = 0.003$ and Tertile 3 versus Tertile 1: RRR = 1.8 (1.2 to 2.8), $p = 0.009$).

Adolescents with the highest decision-making power in Kinshasa had an elevated relative risk of a perpetrator–victim overlap relative to those with the least decision-making power (Boys Tertile 2 versus Tertile 1: RRR = 2.1 (1.3 to 3.2), $p = 0.001$; Boys Tertile 3 versus Tertile 1:

**Table 2. Proportion of peer violence perpetration and/or victimization according to adolescent's level of agency (freedom of movement, voice, and decision-making) by site and gender.**

| | Kinshasa, DRC | | | | | Blantyre, Malawi | | | | |
|---|---|---|---|---|---|---|---|---|---|---|
| | No violence (95% CI) (n) | Victim (95% CI) (n) | Perpetrator | Victim and perpetrator (95% CI) (n) | p-value | No violence (95% CI) (n) | Victim (95% CI) (n) | Perpetrator | Victim and perpetrator (95% CI) (n) | p-value |
| | (95% CI) (n) | | | | | (95% CI) (n) | | | | |
| **Boys** | | | | | | | | | | |
| Freedom of movement | | | | | | | | | | |
| Tertile 1 | 69.0% | 8.5% | 10.0% | 12.5% | | 70.5% | 18.0% | 2.5% | 9.0% | |
| | (64 to 73.4) | (6 to 11.7) | (7.4 to 13.4) | (9.6 to 16.1) | | (61.8 to 77.9) | (12.1 to 25.9) | (0.8 to 7.4) | (5.0 to 15.6) | |
| | (276) | (34) | (40) | (50) | | (86) | (22) | (3) | (11) | |
| Tertile 2 | 65.7% | 10.1% | 9.2% | 15.0% | <0.001 | 47.6% | 25.5% | 10.1% | 16.8% | <0.001 |
| | (60.2 to 70.8) | (7.2 to 14.1) | (6.4 to 12.9) | (11.4 to19.5) (46) | | (40.9 to 54.4) | (20.0 to 31.9) | (6.7 to 15.0) | (12.3 to 22.6) | |
| | (201) | (31) | (28) | | | (99) | (53) | (21) | (35) | |
| Tertile 3 | 50.7% | 13.0% | 12.3% | 24.0% | | 41.4% | 17.2% | 8.2% | 33.2% | |
| | (46.6 to 54.8) | (10.5 to 16.0) | (9.9 to 15.3) | (20.6 to 27.6) | | (35.6 to 47.4) | (13.1 to 22.2) | (5.5 to 12.2) | (27.8 to 39.1) | |
| | (288) | (74) | (70) | (136) | | (111) | (46) | (22) | (89) | |
| Voice | | | | | | | | | | |
| Tertile 1 | 66.3% | 10.9% | 11.7% | 11.2% | | 50.9% | 17.5% | 9.6% | 22.0% | |
| | (61.5 to 70.7) | (8.2 to 14.4) | (8.9 to 15.2) | (8.4 to 14.6) | | (43.5 to 58.2) | (12.6 to 23.9) | (6.0 to 14) | (16.5 to 28.8) | |
| | (267) | (44) | (47) | (15) | | (90) | (31) | (17) | (39) | |
| Tertile 2 | 57.7% | 11.0% | 10.7% | 20.7% | 0.002 | 46.0% | 23.8% | 8.9% | 21.3% | 0.38 |
| | (52 to 62.4) | (8.2 to 14.4) | (8.1 to 14.1) | (17.0 to 24.9) | | (39.3 to 53.0) | (18.4 to 30.1) | (5.7 to 13.7) | (16.2 to 27.5) | |
| | (237) | (45) | (44) | (85) | | (93) | (48) | (18) | (43) | |
| Tertile 3 | 56.7% | 10.9% | 10.2% | 22.2% | | 51.6% | 19.2% | 5.0% | 24.2% | |
| | (52.2 to 61.2) | (8.3 to 14.1) | (7.8 to 13.3) | (18.6 to 26.2) | | (45.0 to 58.2) | (14.5 to 25.0) | (2.8 to 8.9) | (19.0 to 30.3) | |
| | (261) | (50) | (47) | (102) | | (113) | (42) | (11) | (53) | |
| Decision-making | | | | | | | | | | |
| Tertile 1 | 66.4% | 14.9% | 9.6% | 9.1% | | 54.8% | 17.4% | 9.6% | 18.1% | |
| | (61.6 to 70.9) | (11.7 to 18.8) | (7.1 to 12.9) | (6.6 to 12.4) | | (47.2 to 62.3) | (12.4 to 24.0) | (6.0 to 15.2) | (12.9 to 24.7) | |
| | (263) | (59) | (38) | (36) | | (91) | (29) | (16) | (30) | |
| Tertile 2 | 59.4% | 9.1% | 12.1% | 19.4% | <0.001 | 47.7% | 23.2% | 8.3% | 20.8% | 0.15 |
| | (55.4 to 63.4) | (7.0 to 11.7) | (9.6 to 15.0) | (16.4 to 22.9) | | (41.1 to 54.4) | (18.0 to 29.3) | (5.3 to 12.9) | (15.9 to 26.8) | |
| | (340) | (52) | (69) | (111) | | (103) | (50) | (18) | (45) | |
| Tertile 3 | 52.9% | 9.2% | 10.1% | 27.8% | | 47.2% | 19.4% | 5.6% | 27.8% | |

(*Continued*)

**Table 2.** (Continued)

| | Kinshasa, DRC | | | | | Blantyre, Malawi | | | | |
|---|---|---|---|---|---|---|---|---|---|---|
| | No violence (95% CI) (n) | Victim (95% CI) (n) | Perpetrator | Victim and perpetrator (95% CI) (n) | p-value | No violence (95% CI) (n) | Victim (95% CI) (n) | Perpetrator | Victim and perpetrator (95% CI) (n) | p-value |
| | | | (95% CI) (n) | | | | | (95% CI) (n) | | |
| | (47.3 to 58.4) | (6.4 to 12.9) | (7.2 to 14.1) | (23.0 to 33.1) | | (41.6 to 53.9) | (14.7 to 25.3) | (3.2 to 9.5) | (22.2 to 34.2) | |
| | (162) | (28) | (31) | (85) | | (102) | (42) | (12) | (60) | |
| Girls | | | | | | | | | | |
| Freedom of movement | | | | | | | | | | |
| Tertile 1 | 68.7% | 9.4% | 12.0% | 9.9% | | 71.3% | 13.8% | 5.5% | 9.5% | |
| | (64.8 to 72.3) | (7.3 to 12.1) | (9.6 to 14.8) | (7.8 to 12.6) | | (65.4 to 76.5) | (10.0 to 18.6) | (3.3 to 9.1) | (6.4 to 13.7) | |
| | (408) | (56) | (71) | (59) | | (181) | (35) | (14) | (95) | |
| Tertile 2 | 65.5% | 9.4% | 14.9% | 10.2% | 0.85 | 61.4% | 18.7% | 5.4% | 14.5% | <0.001 |
| | (60.6 to 70.1) | (6.9 to 12.8) | (11.7 to 18.8) | (7.5 to 13.6) | | (55.1 to 67.4) | (14.2 to 24.1) | (3.2 to 9.1) | (11.0 to 19.6) | |
| | (251) | (36) | (57) | (39) | | (148) | (45) | (13) | (35) | |
| Tertile 3 | 67.5% | 9.3% | 11.8% | 11.4% | | 39.2% | 25.0% | 9.2% | 26.7% | |
| | (61.8 to 72.6) | (6.5 to 13.3) | (8.5 to 16.0) | (8.2 to 15.6) | | (30.8 to 48.2) | (18.0 to 33.6) | (5.1 to 15.8) | (19.5 to 35.3) | |
| | (195) | (27) | (34) | (33) | | (47) | (30) | (11) | (32) | |
| Voice | | | | | | | | | | |
| Tertile 1 | 65.7% | 10.4% | 12.2% | 11.8% | | 67.1% | 15.3% | 5.6% | 12.0% | |
| | (61.3 to 69.7) | (8.0 to 13.3) | (9.6 to 15.4) | (9.2 to 15.0) | | (60.6 to 73.1) | (11.1 to 20.7) | (3.2 to 9.5) | (8.3 to 17.1) | |
| | (323) | (51) | (60) | (58) | | (145) | (33) | (12) | (26) | |
| Tertile 2 | 65.3% | 9.9% | 15.1% | 9.7% | 0.23 | 52.5% | 24.0% | 7.4% | 16.2% | 0.05 |
| | (60.4 to70.0) | (7.4 to 13.3) | (12.0 to 19.0) | (7.1 to 13.0) | | (45.6 to 59.2) | (18.6 to 30.4) | (4.5 to 11.9) | (11.7 to 21.9 | |
| | (263) | (40) | (61) | (39) | | (107) | (49) | (15) | (33) | |
| Tertile 3 | 72.2% | 7.6% | 11.1% | 9.2% | | 63.6% | 14.4% | 5.6% | 16.4% | |
| | (67.4 to 76.6) | (5.3 to 10.7) | (8.2 to 14) | (6.6 to 12.6) | | (56.6 to 70.1) | (10.1 to 20.0) | (3.1 to 9.9) | (11.8 to 22.3) | |
| | (268) | (28) | (41) | (34) | | (124) | (28) | (11) | (32) | |
| Decision-making | | | | | | | | | | |
| Tertile 1 | 69.2% | 9.0% | 12.6% | 9.2% | | 69.6% | 11.1% | 6.8% | 12.6% | |
| | (64.8 to 73.3) | (6.7 to 12.0) | (9.8 to 16.0) | (6.9 to 12.3) | | (62.9 to 75.5) | (7.5 to 16.2) | (4.0 to 11.1) | (8.7 to 17.8) | |
| | (308) | (40) | (56) | (41) | | (144) | (23) | (14) | (26) | |
| Tertile 2 | 71.0% | 9.5% | 10.5% | 9.0% | 0.04 | 57.1% | 21.9% | 5.6% | 15.5% | 0.04 |
| | (66.7 to 74.9) | (7.1 to 12.5) | (8.1 to 13.7) | (6.7 to 12.0) | | (50.6 to 63.3) | (17.0 to 27.7) | (3.3 to 9.4) | (11.3 to 20.7) | |
| | (330) | (44) | (49) | (42) | | (133) | (51) | (13) | (36) | |
| Tertile 3 | 60.7% | 9.8% | 16.0% | 13.5% | | 56.6% | 20.6% | 6.3% | 16.6% | |

(Continued)

**Table 2.** (Continued)

| Kinshasa, DRC | | | | | Blantyre, Malawi | | | | |
|---|---|---|---|---|---|---|---|---|---|
| No violence (95% CI) (*n*) | Victim (95% CI) (*n*) | Perpetrator | Victim and perpetrator (95% CI) (*n*) | *p*-value | No violence (95% CI) (*n*) | Victim (95% CI) (*n*) | Perpetrator | Victim and perpetrator (95% CI) (*n*) | *p*-value |
| (95% CI) (*n*) | | | | | (95% CI) (*n*) | | | | |
| (55.5 to 65.6) | (7.1 to 13.4) | (12.6 20.2) | (10.3 to 17.5) | | (49.1 to 63.8) | (15.2 to 27.2) | (3.5 to 11.0) | (11.7 to 22.9) | |
| (216) | (35) | (57) | (48) | | (99) | (36) | (11) | (29) | |

Chi-squared tests were used to assess differences in peer violence experiences by agency tertiles.

DRC, Democratic Republic of Congo.

RRR = 3.0 (1.8 to 4.8), *p* < 0.001; and Girls Tertile 3 versus Tertile 1: RRR = 1.7 (1.0 to 2.7), *p* = 0.04). While interpretation is limited given methodological differences by site, we also note significant differences in the associations between freedom of movement and peer violence that are stronger among girls in Blantyre compared to girls in Kinshasa girls (S2 Table), which remained even after excluding out-of-school adolescents in Kinshasa (S3 Table). No site interactions were noted for boys.

Other factors related to peer violence in the multivariate analyses are presented in S4 and S5 Tables. In both sites, older boys were less likely to report being victimized compared to younger boys (in Kinshasa: RRR = 0.6 (0.4 to 0.9), *p* = 0.03 and in Blantyre RRR = 0.4 (0.2 to 0.7), *p* < 0.001)). Likewise, older boys in Kinshasa were less likely to report perpetration–victimization overlap than younger boys (RRR = 0.6 (0.4 to 0.8), *p* = 0.001). Boys who were in a grade lower than age-expected grade were less likely to be victimized compared to out-of-school boys in Kinshasa (RRR = 0.5 (0.2 to 0.9), *p* = 0.03), and boys who were in their age expected grade or higher were less likely to be victimized than boys who were in a grade lower than age-expected grade in Blantyre (RRR = 0.5 (0.3 to 0.97), *p* = 0.04). In Kinshasa, greater wealth was associated with a lower risk of victimization (wealth quintile 2 versus 1: RRR = 0.6 (0.3 to 0.99), *p* = 0.048) and a lower risk of perpetration–victimization overlap (wealth quintile 2 versus 1: RRR = 0.5 (0.3 to 0.8), *p* = 0.005 and wealth quintile 4 versus 1: RRR = 0.4 (0.3 to 0.8), *p* = 0.004) compared to adolescents living in the poorest households. In both sites, adolescents who reported a history of 2 or more ACEs had elevated risk of all forms of peer violence experiences relative to those with no history of ACEs. Parent and peer relations, social cohesion, and social norms were related to peer violence experiences, although these associations differed by site. Among Kinshasa boys, being close to a parent was associated with an elevated relative risk of victimization (RRR = 1.6 (1.04 to 2.4), *p* = 0.03) and a lower risk of perpetration (RRR = 0.7 (0.5 to 0.99), *p* = 0.046) and of perpetration and victimization overlap (RRR = 0.6 (0.4 to 0.8), *p* = 0.001) compared to boys who were not close to their parents. Likewise, girls in Kinshasa who were close to their parents had a lower relative risk of violence perpetration (RRR = 0.5 (0.4 to 0.8), *p* = 0.001) and of perpetration–victimization overlap (RRR = 0.5 (0.3 to 0.8), *p* = 0.002) compared to girls who were not close to their parents. In Blantyre, boys with opposite gender friends had an elevated relative risk of perpetration relative to those with same gender friends (RRR = 3.9 (1.3 to 12.1), *p* = 0.02), while girls with opposite gender friends had an elevated relative risk of perpetration–victimization overlap (RRR = 2.1 (1.04 to 4.3), *p* = 0.04). In Kinshasa, high social cohesion was associated with a lower relative risk of perpetration among boys (RRR = 0.6 (95% CI: 0.4 to 0.9), *p* = 0.02); however, among girls, it was associated with an elevated relative risk of perpetration (RRR = 2.0 (1.3 to 3.0), *p* = 0.001) and of combined perpetration victimization (RRR = 2.1 (95% CI: 1.3 to 3.3), *p* = 0.003). In

**Table 3.** RRRs of violence victimization, violence perpetration, and violence victimization and perpetration by agency indicators: results from multivariate multinomial regression models [1].

| | Kinshasa, DRC | | | Blantyre, Malawi | | |
|---|---|---|---|---|---|---|
| | Victimized RRR (95% CI) (*p*-value) | Perpetrated RRR (95% CI) (*p*-value) | Victimized and perpetrated RRR (95% CI) (*p*-value) | Victimized RRR (95% CI) (*p*-value) | Perpetrated RRR (95% CI) (*p*-value) | Victimized and perpetrated RRR (95% CI) (*p*-value) |
| **Boys** | | | | | | |
| **Freedom of movement** | | | | | | |
| *Tertile 1* | Ref | Ref | Ref | Ref | Ref | Ref |
| *Tertile 2* | 1.5 (0.9 to 2.5) (*p* = 0.17) | 0.9 (0.5 to 1.6) (*p* = 0.74) | 1.1 (0.7 to 1.7) (*p* = 0.73) | 2.1 (1.1 to 3.9) (*p* = 0.02) | 7.2 (2.0 to 26.4) (*p* = 0.003) | 2.4 (1.1 to 5.5) (*p* = 0.03) |
| *Tertile 3* | 2.8 (1.7 to 4.6) (*p* < 0.001) | 1.7 (1.1 to 2.6) (*p* = 0.03) | 1.9 (1.2 to 2.8) (*p* = 0.003) | 1.3 (0.7 to 2.5) (*p* = 0.45) | 5.9 (1.6 to 21.9) (*p* = 0.009) | 3.8 (1.7 to 8.3) (*p* = 0.001) |
| **Voice** | | | | | | |
| *Tertile 1* | Ref | Ref | Ref | Ref | Ref | Ref |
| *Tertile 2* | 1.2 (0.8 to 2.0) (*p* = 0.42) | 1.1 (0.7 to 1.8) (*p* = 0.67) | 1.9 (1.2 to 2.9) (*p* = 0.003) | 1.3 (0.7 to 2.3) (*p* = 0.45) | 0.7 (0.3 to 1.7) (*p* = 0.44) | 0.7 (0.4 to 1.4) (*p* = 0.37) |
| *Tertile 3* | 1.2 (0.7 to 2.0) (*p* = 0.43) | 1.2 (0.7 to 1.9) (*p* = 0.52) | 1.8 (1.2 to 2.8) (*p* = 0.009) | 1.0 (0.5 to 1.9) (*p* = 0.98) | 0.4 (0.1 to 1.2) (*p* = 0.10) | 0.7 (0.4 to 1.4) (*p* = 0.29) |
| **Decision-making** | | | | | | |
| *Tertile 1* | Ref | Ref | Ref | Ref | Ref | Ref |
| *Tertile 2* | 0.6 (0.4 to 0.9) (*p* = 0.007) | 1.3 (0.8 to 2.0) (*p* = 0.30) | 2.1 (1.4 to 3.3) (*p* = 0.001) | 1.5 (0.8 to 2.7) (*p* = 0.20) | 0.9 (0.4 to 2.1) (*p* = 0.83) | 1.2 (0.7 to 2.4) (*p* = 0.51) |
| *Tertile 3* | 0.6 (0.4 to 1.0) (*p* = 0.07) | 1.1 (0.6 to 1.9) (*p* = 0.70) | 3.0 (1.8 to 4.8) (*p* < 0.001) | 1.4 (0.7 to 2.6) (*p* = 0.36) | 0.6 (0.2 to 1.7) (*p* = 0.36) | 1.7 (0.8 to 3.3) (*p* = 0.14) |
| **Girls** | | | | | | |
| **Freedom of movement** | | | | | | |
| *Tertile 1* | Ref | Ref | Ref | Ref | Ref | Ref |
| *Tertile 2* | 1.1 (0.7 to 1.8) (*p* = 0.63) | 1.3 (0.9 to 2.0) (*p* = 0.20) | 1.1 (0.7 to 1.8) (*p* = 0.58) | 1.0 (0.6 to 1.8) (*p* = 0.89) | 0.9 (0.4 to 2.0) (*p* = 0.77) | 1.1 (0.6 to 2.0) (*p* = 0.84) |
| *Tertile 3* | 1.0 (0.6 to 1.8) (*p* = 0.88) | 0.9 (0.5 to 1.4) (*p* = 0.54) | 1.1 (0.7 to 1.8) (*p* = 0.74) | 1.9 (0.9 to 3.8) (*p* = 0.08) | 2.5 (0.9 to 6.8) (*p* = 0.08) | 2.4 (1.1 to 5.1) (*p* = 0.03) |
| **Voice** | | | | | | |
| *Tertile 1* | Ref | Ref | Ref | Ref | Ref | Ref |
| *Tertile 2* | 0.8 (0.5 to 1.3) (*p* = 0.35) | 1.2 (0.8 to 1.9) (*p* = 0.38) | 0.8 (0.5 to 1.3) (*p* = 0.32) | 1.9 (1.1 to 3.1) (*p* = 0.02) | 1.8 (0.8 to 4.5) (*p* = 0.18) | 2.0 (1.01 to 3.9) (*p* = 0.048) |
| *Tertile 3* | 0.6 (0.3 to 1.0) (*p* = 0.052) | 0.8 (0.5 to 1.3) (*p* = 0.30) | 0.6 (0.4 to 1.0) (*p* = 0.07) | 0.8 (0.4 to 1.5) (*p* = 0.47) | 1.0 (0.4 to 2.7) (*p* = 0.98) | 1.3 (0.7 to 2.7) (*p* = 0.40) |
| **Decision-making** | | | | | | |
| *Tertile 1* | Ref | Ref | Ref | Ref | Ref | Ref |
| *Tertile 2* | 1.1 (0.7 to 1.7) (*p* = 0.81) | 0.7 (0.5 to 1.1) (*p* = 017) | 0.9 (0.5 to 1.4) (*p* = 0.61) | 1.8 (1.0 to 3.2) (*p* = 0.06) | 0.8 (0.3 to 1.8) (*p* = 0.52) | 0.8 (0.4 to 1.6) (*p* = 0.59) |
| *Tertile 3* | 1.5 (0.9 to 2.6) (*p* = 0.14) | 1.3 (0.9 to 2.1) (*p* = 0.18) | 1.7 (1.02 to 2.7) (*p* = 0.04) | 1.7 (0.9 to 3.3) (*p* = 0.1) | 0.8 (0.3 to 2.1) (*p* = 0.68) | 1.0 (0.5 to 2.0) (*p* = 0.94) |

[1] All models estimate the relative risk of violence victimization of violence perpetration and of violence perpetration–victimization overlap relative to no violence according to tertiles of agency after adjusting for all other covariates (individual, family, peer, and community factors).

DRC, Democratic Republic of Congo; RRR, relative risk ratio.

Blantyre, high social cohesion among boys was associated with an elevated relative risk of victimization (RRR = 1.7 (1.03 to 2.8), *p* = 0.04) and of a victimization and perpetration overlap (RRR = 3.2 (1.8 to 5.7), *p* < 0.001). Finally, higher perception of gender stereotypical traits was

associated with an elevated relative risk of a victimization and perpetration overlap among girls in Kinshasa (RRR = 1.7 (95% CI: 1.1 to 2.6), *p* = 0.01).

## Discussion

This study adds to the scarce literature exploring peer violence and individual agency among VYAs living in poor urban communities in two low-income countries. Boys in both settings were more likely to experience peer violence than girls, mostly due to higher rates of a perpetration and victimization overlap. Agency had different connections to peer violence experiences, by gender and site, although greater freedom of movement, voice, and decision-making power were generally related to increases in some form of peer violence experience. These associations were generally greater for boys than girls and often showed different associations in relation to victimization versus perpetration, when they occurred in isolation of each other.

The gender divide in peer violence exposure is well described in previous literature, reporting 3 to 12 times higher prevalence among boys compared to girls [9]. While mostly described in the form of female victimization and male perpetration [33,34], our results suggest more complex gender dynamics, as differences mostly originated from a greater perpetrator–victim overlap among boys than girls. These results are supported by other studies in Pakistan and Afghanistan indicating a larger perpetrator–victim overlap among boys than girls [35,36], suggesting boys may be more likely than girls to retaliate when victimized. Our study found that in Kinshasa, girls who had unequal perceptions of gender stereotypical traits, promoting male toughness over female vulnerability, were more likely to report both perpetration and victimization, highlighting the connection between gender norms and peer violence experiences in this age group. The fact that this was only true among girls in Kinshasa illustrates the fact that gender norms have different meanings across contexts and influence behaviors in different ways for boys and girls [30].

Gender differences in peer violence experiences partly stem from a gender divide in freedom of movement. Boys enjoy greater freedom of movement than girls and spend more time socializing with their friends [28,37,38], due to normative expectations, prescribing more protection for girls and more independence for boys [21,30]. In addition, freedom of movement was more strongly related to violence experiences among boys than girls. A study of low-income Latino adolescents in two cities in the United States showed that more unstructured time with peers increased community violence and victimization among boys [39]. Together, these findings draw attention to gender differences in social environments and activities outside of the home contributing to greater male exposure and engagement in violence.

Decision-making power and voice, which are expressions of empowerment, are associated with reductions in gender-based violence against women, although greater empowerment may also lead to social backlash resulting in violence victimization [17]. In our study, we also suggest a complex relation between decision-making power, voice, and peer violence experiences, which differs by gender and site. For example, in Kinshasa, greater decision-making power was associated with lower risk of victimization for boys but not for girls. Greater voice among girls was associated with increased risk of peer violence in Blantyre but not in Kinshasa. These variations of effects have been described in a systematic review of interventions to prevent gender-based violence among adolescent girls, showing positive, null, or even negative effects of raising girls' voice and agency in preventing violence against women, depending on context [17].

Across all groups, a history of childhood adverse events was associated with increased risk of violence victimization, perpetration, or both. These results are consistent with prior studies conducted in high-income and low-income settings, including studies using the current GEAS

measure [40,41]. Other contextual factors related to parent and peer relations were also associated with young people's violence experience, although these relations differed by site. Literature has shown that both family and peer factors affect adolescent violent behaviors [10,11].

Our results need to be interpreted with some limitations in mind. The cross-sectional design of this study limits our interpretation of the perpetrator–victim overlap, as well as the directionality of the associations described. Although we assume that participants with more agency are more prone to peer violence, it could be that those who engage in peer violence have more agency. While our results demonstrate that agency and peer violence are associated, we anticipate that the longitudinal data currently collected from the GEAS will provide further insights on the dynamics of peer violence experiences (from victimization to perpetration and vice versa) as well as the predictive effects of young people's agency on subsequent violence outcomes. We also acknowledge the limitation of our peer violence measure, which only focuses on physical violence and does not account for gender-based violence behaviors. The small sample sizes prevented more refined analysis of same gender versus opposite gender violence experiences, and additional information would be needed to consider other types of peer violence (verbal, emotional, and cyber), which may also vary by gender and context. We also recognize the potential for residual confounding due to unmeasured confounders in our analysis, particularly as we investigated individual components of adolescent agency but were unable to assess the role of collective empowerment, a critical component of violence prevention programs for teens. Selection bias may also have impacted our findings, particularly if adolescents who are victims of violence or who have limited agency are less likely to be identified and enrolled in the study. If our study differentially selected participants who were less likely to experience violence, it is likely that our findings are conservative. Differences in data collection modalities between sites (in person in Kinshasa and Audio Computer-Assisted Self-Interview in Blantyre) may generate differences in the patterns of responses resulting from modality rather than true differences between sites. This is especially notable in Blantyre, which had a larger number of missing variables compared to Kinshasa. On the other hand, social desirability bias may have impacted responses in Kinshasa collected face to face [42]. We note, however, that violence perpetration was equally prevalent in Blantyre and Kinshasa, which may signal low social desirability bias. Another limitation is that we have little information about the context in which peer violence occurs, which is not only programmatically relevant for violence prevention, but may also inform the relationship between agency and peer violence. Future qualitative work could shed more light on the ways agency and violence interconnect, according to social context. Last, a purposive sample might not affect internal validity but affects the generalizability of our findings for the city and country.

Despite these limitations, this cross-cultural analysis allows a better understanding of peer violence experiences among boys and girls in early adolescence and the complex role of adolescent's agency in shaping these experiences. Our results have important programmatic implications, raising the possibility that gender socialization may expose boys to more interpersonal violence as they enjoy more freedom of movement and are expected to protect themselves. They also draw attention to the potential for unforeseen consequences of empowerment, which was sometimes associated with increased peer violence experiences in our study. We suggest the following programmatic implications. First, programs that promote young people's agency need to conjointly address gender norms and violence as normative expectations may lead those who gain power toward increased aggression as a means of assertion and self-protection. Second, the increase in violence perpetration and victimization among adolescents who have experienced ACEs as well as the extent of a perpetration and victimization overlap reaffirms the importance of a trauma-informed care to prevent teen violence. Third, strategies to prevent youth violence need to engage the broader social environment, starting with the

family and extending to communities, to reduce violence exposure and perpetration. For instance, while many parents recognize the risk of violence for their sons, many fail to act considering that their sons need to rely on their own agency to protect themselves [21]. The role of the social context, illustrated in the differences in association by site, also stresses the importance of including young people and their social networks in the design and implementation of interventions in order to better address the social and structural drivers of teen violence. Training community members would ensure that interventions are trusted, followed, and sustained.

## Supporting information

**S1 STROBE Checklist. STROBE guideline checklist.** STROBE, Strengthening the Reporting of Observational Studies in Epidemiology.
(DOCX)

**S1 IRB. Baseline IRB Protocol for Kinshasa, DRC.** DRC, Democratic Republic of Congo; IRB, Institutional Review Board.
(DOCX)

**S2 IRB. Baseline IRB Protocol for Blantyre, Malawi.** IRB, Institutional Review Board.
(DOCX)

**S1 Table. GEAS questions and measures.** GEAS, Global Early Adolescent Study.
(DOCX)

**S2 Table. Differences in associations between agency and peer violence by site among adolescent boys and girls: results from pooled multivariate multinomial regression model across sites, including interaction terms by site.**
(DOCX)

**S3 Table. Overall model with interaction between site and agency variables and excluding out-of-school participants.**
(DOCX)

**S4 Table. Factors related to peer violence among boys and girls in Kinshasa: results from multivariate multinomial regression model.**
(DOCX)

**S5 Table. Factors related to peer violence among boys and girls in Malawi: results from multivariate multinomial regression model.**
(DOCX)

**S1 Data. Dataset used for analysis.**
(CSV)

**S1 Codebook. Codebook of the variables used for analysis.**
(PDF)

## Author Contributions

**Conceptualization:** Astha Ramaiya, Caroline Moreau.

**Formal analysis:** Astha Ramaiya.

**Investigation:** Caroline Moreau.

**Methodology:** Astha Ramaiya, Linnea Zimmerman, Caroline Moreau.

**Supervision:** Caroline Moreau.

**Writing – original draft:** Astha Ramaiya, Caroline Moreau.

**Writing – review & editing:** Astha Ramaiya, Linnea Zimmerman, Eric Mafuta, Aimee Lulebo, Effie Chipeta, William Stones, Caroline Moreau.

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
