## [Editor Report · Decision Letter 0]

16 Feb 2021

Dear Dr Ramaiya, 

Thank you for submitting your manuscript entitled "Assessing the Relationship Between Agency and Peer-violence among Very Young Adolescents in Two African Settings: A Cross-Sectional Study" for consideration in PLOS Medicine’s Special Issue on Global Child Health.

Your manuscript has now been evaluated by the PLOS Medicine editorial staff as well as by the Special Issue Guest Editors, and I am writing to let you know that we would like to send your submission out for external peer review.

Kind regards,

Caitlin Moyer, Ph.D.

Associate Editor

PLOS Medicine

---

## [Decision Letter · Decision Letter 1]

2 Apr 2021

Dear Dr. Ramaiya,

Thank you very much for submitting your manuscript "Assessing the Relationship Between Agency and Peer-violence among Very Young Adolescents in Two African Settings: A Cross-Sectional Study" (PMEDICINE-D-21-00483R1) for consideration in PLOS Medicine’s Special Issue on Global Child Health.

Your paper was evaluated by a senior editor and discussed among all the editors here. It was also discussed with the special issue guest editors, and sent to three independent reviewers, including a statistical reviewer. The reviews are appended at the bottom of this email and any accompanying reviewer attachments can be seen via the link below:

[LINK]

In light of these reviews, I am afraid that we will not be able to accept the manuscript for publication in the journal in its current form, but we would like to consider a revised version that addresses the reviewers' and editors' comments. Obviously we cannot make any decision about publication until we have seen the revised manuscript and your response, and we plan to seek re-review by one or more of the reviewers. 

We expect to receive your revised manuscript by . Please email us (plosmedicine@plos.org) if you have any questions or concerns.

We look forward to receiving your revised manuscript. 

Sincerely,

Caitlin Moyer, Ph.D.

Associate Editor 

PLOS Medicine

plosmedicine.org

1. Data availability statement: Thank you for noting the specific restrictions for access of your data. Please revise your statement, to include appropriate information allowing interested parties to access your study data (a dataset reference or accession number, a web link, and/or contact email addresses). Please note that the authors of the study (Eric Mafuta and William Stones) may not serve as contact points for providing data access. 

2. When submitting your revised manuscript, please provide line numbers throughout the document.

3. Abstract: Please structure your abstract using the PLOS Medicine headings (Background, Methods and Findings, Conclusions; please combine methods and results into a single section “Methods and Findings”).

4. Abstract: Background: Please indicate Blantyre, Malawi,

5. Abstract: Methods and Findings: Please quantify the main results (with 95% CIs and p values).

6. Abstract: Methods and Findings: In the last sentence of the Abstract Methods and Findings section, please describe the main limitation(s) of the study's methodology.

7. Abstract: Conclusions: In the first sentence please address the specific implications of your study based on the results; the phrase "In this study, we observed ..." may be useful.

8. Author summary: At this stage, we ask that you include a short, non-technical Author Summary of your research to make findings accessible to a wide audience that includes both scientists and non-scientists. The Author Summary should immediately follow the Abstract in your revised manuscript. This text is subject to editorial change and should be distinct from the scientific abstract. Please see our author guidelines for more information: https://journals.plos.org/plosmedicine/s/revising-your-manuscript#loc-author-summary

9. Throughout the text: Please use square brackets for in-text citations, like this [1].

10. Methods: Reporting Checklist: Please ensure that the study is reported according to the STROBE guideline, and include the completed STROBE checklist as Supporting Information. When completing the checklist, please use section and paragraph numbers, rather than page numbers. Please add the following statement, or similar, to the Methods: "This study is reported as per the Strengthening the Reporting of Observational Studies in Epidemiology (STROBE) guideline (S1 Checklist)." 

11. Methods: Analysis Plan: Did your study have a prospective protocol or analysis plan? Please state this (either way) early in the Methods section.

12. Methods: Page 7: Please remove the trademark symbol from SurveyCTO®

13. Methods: Page 7: Please provide more detail on whether the description here applies to both child assent and parental consent, or whether these differed.

14. Methods: Please include the survey questions used to assess these three agency dimensions, and all covariates, as a supporting information file.

15. Methods: Page 9: Please clarify the sentence: “We first examined the incidence of violence perpetration and victimization in the last § months…”

16. Results: Please quantify results presented in the text as much as possible. On page 14, please present the risk ratio in the text along with the results from the multivariate analysis (95% CI and p values) for the findings discussed. Please clarify where differences between sites or sexes were quantified, as opposed to where differences are apparent at the qualitative level.

17. Discussion: Please slightly re-organize the Discussion as follows: a short, clear summary of the article's findings; what the study adds to existing research and where and why the results may differ from previous research; strengths and limitations of the study; implications and next steps for research, clinical practice, and/or public policy; one-paragraph conclusion.

18. Discussion and throughout: It is somewhat confusing to switch back and forth between “gender” and “sex” (even within a single sentence at the top of page 18). I understand the point you make on page 9 that sex is the relevant term in your study as you did not collect data on gender identity, and that your survey contains gender-related measures. Please make sure the two terms (which have different meaning) are used as consistently and clearly as possible throughout.

19. Discussion: In the limitations, please fully discuss limitations of the study design, including selection bias brought up by one reviewer, as well as issues related to residual confounding.

20. Table 1: Please provide the number for each percentage given (even though the total is given at the top of the columns). Please define the abbreviation ACES, SD, in the legend.

21. Figure 1: Please consider that this may be more clearly presented as a table. Please also provide the number in addition to the percentages for each value.

22. Table 2: Please provide the numbers for each percentage. Please report the full statistics for each comparison, along with the 95% CIs and p values, rather than noting significance with asterisks.

23. Table 3: Please report the full p values for each comparison, rather than indicating significance levels with asterisks.

24. References: Please use the "Vancouver" style for reference formatting, and see our website for other reference guidelines https://journals.plos.org/plosmedicine/s/submission-guidelines#loc-references

25. Appendix 1 and Appendix 2: Please provide the actual p values associated with the results presented, rather than levels of significance. In the legend, please briefly explain why parts of the table are grayed out. We suggest renaming these files as “S1_Table” and “S2_Table” respectively.

Comments from the reviewers:

Reviewer #1: This paper addresses the important topic of interpersonal violence among adolescents in Kinshasa and Blantyre. Overall, the study suffers from several limitations related to its cross sectional design, including the potential for reverse causation and selection bias. Moreover, the target population for the study is not explicitly stated, and the method used to sample, invite, and recruit participants was not clear. Therefore, it is difficult to understand to whom results are expected to apply. General and specific comments are provided below.

General comments: 

1. The paper does not describe how participants were selected for inclusion. For example, how were the 2016 in school and 826 out of school adolescents in Kinshasa selected? How many were invited to participate? What was the refusal rate?

2. Are there circumstances in which experiences of violence would affect participation in the survey? Certainly this could be true in the unfortunate event that violence led to death. But violence could also affect school drop out, which could affect inclusion in the study in Malawi, and may affect recruitment in other ways. Moreover, agency (freedom of movement, voice, and decision-making) could also affect inclusion in the study. In settings where both exposure and outcome affect selection, the overall prevalence of reported outcomes as well as the associations between exposure and outcome could be distorted. This potential selection bias should be discussed.

3. The authors should comment on whether they believe that the difference in data collection techniques (ACASI vs in person interviews) could result in differences in results between the two countries.

4. The Methods section should clearly state the parameters of interest that will be interpreted in the Results section. 

5. The paper acknowledges the limitations of working with cross sectional data in the Discussion section. However, more discussion is needed on the potential for reverse causation (i.e., in which violence affects the agency measures) and what that might mean for the interpretation of the results.

6. The items used to construct measures of agency should be described either in the text or supplementary materials.

Specific Comments: 

1. Abstract: the first sentence of the methods states that the surveys were conducted in 2017 and 2020, yet the methods section states that data collection occurred between 2017 and 2020. I recommend providing more detail on specific dates of data collection in each site.

2. Page 9, last paragraph: the manuscript uses a symbol I don't understand here. What was the lookback period for incidence of violence?

3. Table 2: I appreciate that the paper shows absolute percentages experiencing each level of violence by tertile. However, because there are many categories, it is unclear what hypothesis the p-value is testing and what test was used to obtain the p-value. These point should be clarified if p-values are reported.

4. Page 13, first sentence: Table 2 appears to show absolute percentages of each tertile in each category of the outcome, not associations directly.

5. Page 14, first sentence: it is unclear to me what the term "relative risk ratio" represents here. Presumably the results of Table 3 come from the multiviariable multinomial logistic regression model mentioned in the Methods. However, that method would provide odds ratios, rather than risk ratios. If the "relative risk ratios" are calculated using some other approach, it should be detailed in the Methods section.

6. Moreover, to avoid confusion with "multivariate" models (typically used when more than one outcome is of interest), I would refer to the analysis in table 3 as the "adjusted analysis" rather than "multivariate analysis".

7. Page 14, second paragraph: the paper appears to commit the Table 2 fallacy, in which a multivariable model was used to assess, and interpret, relationships between all variables included in the model and the outcome. However, using results from a single multivariable model, one should not interpret all of the resulting regression coefficients because many of the characteristics affect, and are affected by, each other. This means that the interpretation of some resulting regression coefficients (or odds ratios) would be as "total effects" and others would be "direct effects after holding mediators constant" This phenomenon is nicely described in the reference below.

Daniel Westreich, Sander Greenland; The Table 2 Fallacy: Presenting and Interpreting Confounder and Modifier Coefficients, American Journal of Epidemiology, Volume 177, Issue 4, 15 February 2013, Pages 292-298, https://doi.org/10.1093/aje/kws412

8. Table 3, in the methods section and/or the footnote of the table, please state what p-values are comparing and how they were obtained.

9. Figure 1 was a bit difficult to read, I recommend larger text.

Reviewer #2: Review of Assessing the Relationship Between Agency and Peer-violence among Very Young Adolescents in Two African Settings: A Cross-Sectional Study

Overall, this is an interesting and competently written paper focusing on an important and often neglected age cohort - very young adolescents - in two urban populations in SSA. It makes the important argument that sex differences in the perpetration or victimisation of peer violence vary across context and to support interventions more precise understanding and operationalisation of individual adolescent empowerment is required. 

In some ways, however, the article raises more questions than it answers and either additional analysis should be undertaken to address these or the limitations section expanded. 

First, it would be good if the article could more clearly state whether peer violence is referring to violence experiences with young people only of the same sex or among girls and boys, and if so what proportion. For example, is the analysis of girls' experience of peer violence capturing violence by girls against girls or by boys or both and what is the balance of this experience. This matters in terms of the sorts of sensitisation and monitoring required. 

Second, in what spaces is the violence occurring. The paper makes the argument that greater freedom of movement permitted of boys means that they are more exposed to peer violence experiences - but where is the violence happening? At school or in the community? And do we see differences in the Kinshasa sample based on school enrolment status? 

Third, there are number of areas where the reader wants to know why. With only survey data -rather than mixed methods data (although my understanding is that GEAS does also collect qualitative data) - it is difficult to get purchase on for example why were sex differences in the perpetration or victimisation alone negligible? Why did girls with greater voice also find themselves still at risk of victimisation by peers -was this due to inter-personal communication skills (e.g. limited negotiation skills, gender norms that de-value girls' voices, participation in programming that focused on young people's empowerment but did not foster change in the wider enabling environment), gender-based harassment and violence, and so forth? If it is not possible to draw on qualitative data to triangulate the findings for this paper, then it would be good to add this to the limitations section. 

Fourth, given the sensitivity of asking about experiences of violence, it would be important for the authors to reflect more on possible limitations of different survey data collection methods (i.e. with trained enumerators in Kinshasa vs self-reported in Blantyre). Could this potentially explain some of the level of difference across country? 

Fifth, given that in Kinshasa there are both in-school and out-of-school adolescents, to what extent does this explain differences across country? Could the data be dis-aggregated and bar graphs added in to show the findings for these different samples? If not, it needs to be explicitly discussed in the limitations. Also the descriptive statistics show that there is a difference among those who at-age-for-grade and those who are over-age-for-grade - can the data be dis-aggregated to explore possible differences in findings among these sub-groups? 

Finally, the discussion on implications for programming are quite general and there is a need for more nuanced discussion on how the proposed recommendations could be realistically achieved in such contexts. For example, "Second, while empowering young people is a critical component of positive youth development, promoting individual agency may also have unforeseen consequences, which should be anticipated and addressed to reduce violence experiences". As a reader I would like to know how this could be achieved and whether there are any examples of promising practice in this regard. Similarly, for this recommendation "Finally, the site differences in associations according to peer, family and neighborhood-context stress the need to ground interventions in the social context of young people's lives." - what does this mean in practice - more tailored case management? 

With these revisions i am of the view that this paper can make a strong contribution to the literature on adolescent peer violence. 

Reviewer #3: Referee Report: Assessing the relationship between agency and peer-violence among Very Young Adolescents in Two African Settings: A Cross-Sectional Study

1. Summary

This paper uses cross-sectional data on very young adolescents from two urban cities (Kinshasa (DRC) and Blantyre (Malawi) to assess the association between agency (operationalized with three scales: freedom from movement, voice, and decision making) and peer violence (defined as four categorical variables: no victimization/perpetration, victimization, perpetration, both victimization and perpetration). Analyzing results by sex and site, the authors find that adolescents reporting greater freedom of movement were more likely to experience offender-victim overlap, with more complicated associations for decision making and voice.

2. Major Comments

The paper was very interesting to read, and I appreciate the focus on defining peer violence looking at both victimization and perpetration (and the combination) with a gender lens. It also highlights the complex relationship between agency and peer violence and provides food for thought on the possible adverse consequences of programs that promote agency, without considering the environments adolescents live in.

I do think the authors could do more to motivate the research question, ensure the empirical analysis reflects the complexity laid out in the motivation, and to build more on the programmatic implications. Some specific suggestions below.

a. I am still not sure what to make of the research question. The introduction does a good job motivating the issue of peer violence and the issue of agency, but less on the association between the two. Why would we expect a relationship between the two in LMICs? And why is this likely to be gendered? What evidence do we already have on this association? If agency puts you at risk of exposure to violence, what does this mean for policy etc.? This comes out a bit clearer in the discussion, but I think the introduction could provide clearer motivation of the evidence base and what this paper hopes to add.

b. The research question is motivated within a socioecological model, and I am assuming Table 2 controls for all the variables that are in the appendix tables, is that correct? It would be great to unpack this relationship more, and perhaps explore moderators/mediators more explicitly. I personally think those appendix tables are important and would add value to the main text. One way to build on what is here is instead to show Table 1 as a regression (with no additional controls) and then add controls at different levels. Theory/existing evidence may also suggest the possibility of interacting the measures of agency with other relevant measures. Given the richness of the data, it seems like some additional analysis might shed further light on the complex findings. 

c. The decision to create tertiles of the three dimensions of agency appears somewhat arbitrary. Is there a literature to justify this treatment of the data? Do results look similar if you have a continuous variable or above/below median? Are these tertiles created within gender/site? If you continue to use tertiles, the data in Table 1 should also be summarized this way. It would be good to know the mean (s.d.) freedom of movement for each tertile etc.

d. Are the differences between boys and girls significant? Are the differences across locations significant? Given that one site has only school kids and the other has both, does that explain the difference in findings across contexts?

3. Minor Comments

* Given that GEAS covers a lot more countries, and now has longitudinal data for some, why limit to these two countries in the cross-section? It also seems like a bid of an odd choice given the very different sampling frames for the two contexts. Perhaps only focusing on in school adolescents in the two contexts would allow for a clearer comparison? I am sure the authors have their reasons, but this needs at least a footnote to justify the choice. 

* I suggest adding the data from Figure 1 to Table 1, the figure does not add that much value.

[LINK]

---

## [Decision Letter · Decision Letter 2]

14 Oct 2021

Dear Dr. Ramaiya,

Thank you very much for submitting your revised manuscript "Assessing the Relationship Between Agency and Peer-violence among Very Young Adolescents in Two African Settings: A Cross-Sectional Study" (PMEDICINE-D-21-00483R2) for consideration in PLOS Medicine’s Special Issue: Global Child Health: From Birth to Adolescence and Beyond.

Your paper was evaluated by a senior editor and discussed among all the editors here. It was also discussed with an academic editor with relevant expertise, and re-evaluated by one of the original reviewers. The reviews are appended at the bottom of this email and any accompanying reviewer attachments can be seen via the link below:

[LINK]

In light of these reviews, I am afraid that we will not be able to accept the manuscript for publication in the journal in its current form. The academic editor has noted several issues requiring further clarification, and we request clarification and more thorough description of the methods and analyses.

We will consider a revised version that addresses the reviewers' and editors' comments. Obviously we cannot make any decision about publication until we have seen the revised manuscript and your response, and we may seek re-review by one or more of the reviewers. 

We expect to receive your revised manuscript by Nov 04 2021 11:59PM. Please email us (plosmedicine@plos.org) if you have any questions or concerns.

We look forward to receiving your revised manuscript. 

Sincerely,

Caitlin Moyer, Ph.D.

Associate Editor

PLOS Medicine

plosmedicine.org

1. From the Academic Editor: Please address the following comments:

-The measure of victimization/perpetration is based on only 2 questions focused on physical perpetration/victimization. This type of behavior may be gender-specific, other types of peer-violence (e.g. relational) may be gender-specific as well.

- The temporal ordering between outcome (over the prior 6 months) and primary explanatory variables (agency, at the time of survey) is problematic in the context of the cross-sectional nature of the data.

-The description of agency-related variables is limited and warrants more detail.

-Sex differences are referred to and it seems as if these should be referred to as gender differences.

-The discussion of the analytical strategy is limited, and it is not clear how the nested nature of the data was handled.

Other editorial points:

2. Throughout: Sex vs Gender terminology: It seems as if “sex” was established via the GEAS questions and measure (Supporting File 4) question: “Are you a boy/girl” and we suggest changing “sex” to “gender” throughout. Please also see, https://www.who.int/health-topics/gender.

3. Abstract: Background: It might be helpful to include a sentence briefly describing personal agency.

4. Abstract: Methods and Findings: Please include the percent breakdown by gender for each survey site.

5. Abstract: Methods and Findings: Please report exact p values, unless p<0.001.

6. Abstract: Methods and Findings: Please clarify that T represents tertile at first use of the abbreviation.

7. Abstract: Line 60-61: We suggest revising to “The main limitation of the study is that due to the cross-sectional nature…”

8. Author summary: Line 85-86: Here, and throughout the manuscript, please avoid language implying causality: “...higher agency resulted in higher likelihood of being in the perpetrator-victim overlap group…”

9. Introduction: Line 135-140: Please clarify the final sentence of this paragraph. You mention that empowerment of women/girls is a mechanism to reduce violence against women and challenge hegemonic forms of masculinity, but in the final sentence mention that empowerment programs could backfire in threatening male hegemony in relationships.

10. Methods: Line 174-175: Please provide some more detail on how participants were selected, including selection of schools, participants within schools, participants not enrolled in school.

11. Methods: Line 199-200: Please note that this was parental consent and adolescent assent.

12. Methods: Line 228: Wealth quintile is reported in the S5 and S6 Tables, but the assessment of wealth is not described in the Methods.

13. Methods: Line 259-267: Please consider if the description of your study hypotheses would be better presented in the last paragraph of the Introduction.

14. Results: Please report the numbers of participants at each site, and proportions by gender. present numerators and denominators for percentages.

15. Results: Line 281: Please clarify if this should be boys/men and girls/women.

16. Results: The table shows the overall difference for male vs female in terms of victimization/perpetration but comparisons for perpetration alone or victimiazation alone, for example, are not shown. Please present these: “Sex-differences in perpetration or victimization alone were negligible.”

17. Results: Line 286-290: Please present the results on sex differences in perpetration/victimization (same sex/opposite sex) in a table (such as the one presented in the response to reviewer comments).

18. Results: Line 299-311: Please clarify if the p values reported represent the overall chi-square test (including all three tertiles) or if the results reported reflect comparisons between T1 and T3 specifically. Please describe this analysis more completely in the methods, including whether correction for multiple testing was applied.

19. Results: Line 334-343: Please present the complete results (this can be in a table) for tests done to examine the main effect of site and interactions.

20. Results: Line 342-343: Please present the results of the analyses with out of school students excluded: “These differences remained after excluding out of school adolescent in Kinshasa.”

21. Results: Line 344-361: Please quantify the results in the text, if possible. Please avoid the use of language implying causality in this section (e.g. “...high parental connectedness relative to low connectedness increased the RRR…”

22. Discussion: Line 440: Please expand on the statement that social desirability bias may have factored in to responses from Kinshasa.

23. Table 1: Please note in the legend the statistical tests use to investigate sex differences.

24. Table 2: Please adjust the formatting of the table, as the column widths lead to the text being difficult to read. Please note in the legend the statistical tests used.

25. Table S5 and S6: Instead of using gray boxes, we suggest using different symbols with footnotes to describe the specific reasons why data were unavailable for the analyses.

26. STROBE checklist: Thank you for including the checklist. Please use section and paragraph numbers to identify locations within the text, rather than page numbers.

Comments from the reviewers:

Reviewer #1: Overall, this revision is very responsive to my previous comments. A few issues regarding clarity remain:

In my earlier comments, I was not clear enough about what I meant by "clarifying the Parameters of interest". In the statistical methods section, I recommend that the authors tell the readers the parameter, or estimand, that they will be using and how to interpret it. For example, the main results in table 3 consist of the "relative risk ratio", which, not being a Stata user myself, is an unfamiliar term to me. This measure ought to be defined in the statistical methods section and interpreted for the readers (beyond stating that the relative risk ratio is a measure that comes out of such and such model). What does it mean? If standard multinomial logistic regression is used, this is the odds ratio for being the dependent variable being in the given category rather than the reference category for a one-unit change in the independent variable.

[LINK]

---

## [Editor Report · Decision Letter 3]

18 Nov 2021

Dear Dr. Ramaiya,

Thank you very much for re-submitting your manuscript "Assessing the Relationship Between Agency and Peer-violence among Very Young Adolescents in Two African Settings: A Cross-Sectional Study" (PMEDICINE-D-21-00483R3) for consideration in PLOS Medicine’s Special Issue: Global Child Health: From Birth to Adolescence and Beyond.

I have discussed the paper with my colleagues and the academic editor. I am pleased to say that provided the remaining editorial and production issues are dealt with we are planning to accept the paper for publication in the journal.

[LINK]

We look forward to receiving the revised manuscript by Nov 25 2021 11:59PM.   

Sincerely,

Caitlin Moyer, Ph.D.

Associate Editor 

PLOS Medicine

plosmedicine.org

Requests from Editors:

1. Title: Please revise to include the age range and please use sentence-case capitalization: “Assessing the relationship between agency and peer-violence among adolescents aged 10-14 years in Kinshasa, Democratic Republic of Congo and Blantyre, Malawi: A Cross-Sectional Study”

2. Throughout: Please replace “sex” with “gender” where appropriate, including in the Tables.

3. Abstract: Line 44: Please note that covariates adjusted for include individual, family, peer, and community factors.

4. Author summary: Line 101: We suggest revising to: “both genders” in this point.

5. Author summary: Line 107: Please fully spell out the abbreviation “ACEs” at first use in the text.

6. Introduction: Line 132: Please clarify the direction of relationship between social skills and risk of violence in this sentence.

7. Methods: Line 234, 264: Please change “sex” to “gender” in these sentences.

8. Results and Tables: Please report p values to two decimal places if p>0.01, three decimal places if p>0.001, and for smaller values please report p<0.001.

9. Results: Line 315 and 400-401: Please change “sex” to “gender” if accurate.

10. Results: Line 381-382: Please clarify if this should be “Boys who were in a grade lower than their age-expected grade…”

11. Discussion: Line 478: Please change “sex” to “gender” in this sentence.

12. Discussion: Line 503: We suggest revising to: “...raising the possibility that gender socialization may expose boys to more interpersonal violence as they enjoy more freedom of movement…” or similar.

13. References: Please check each citation for completeness and accuracy. For example, please provide the complete information for Reference 7. For guidance, please use to the "Vancouver" style for reference formatting, and see our website for other reference guidelines https://journals.plos.org/plosmedicine/s/submission-guidelines#loc-references

14. Table 1: Please revise “Peer sex composition” and “Any opposite sex friends” to use “gender” if accurate to report this way.

15. Table 1: Only one category of “Time spent with friends” is reported, though there are 5 categories for this question, please note for this category that responses were collapsed (e.g. please clarify this in the table legend, and in the Methods at line 265). Please also clarify this for “parent connectedness” and any other questions for which responses were collapsed/dichotomized.

16. STROBE Checklist: Please revise the paragraph, using section and paragraph numbers to refer to locations in the text. Please do not refer to page numbers (e.g. Methods, paragraph 2). For “Funding” please refer to the “Financial Disclosures” section.

17. Supplement 9: Thank you for including the data underlying the study as a supporting information file. Please note that we do not require that authors submit their entire data set if only a portion of the data was used in the reported study. The included file contains a large number of variables. A file explaining the nature of each variable would be help researchers to use the dataset.

Prior to sharing human research participant data, authors should consult with an ethics committee to ensure data are shared in accordance with participant consent and all applicable local laws.

Data sharing should never compromise participant privacy. It is therefore not appropriate to publicly share personally identifiable data on human research participants. The following are examples of data that should not be shared: Name, initials, physical address; Internet protocol (IP) address; Specific dates (birth dates, death dates, examination dates, etc.); Contact information such as phone number or email address; Location data.

Steps necessary to protect privacy may include de-identifying data, adding noise, or blocking portions of the database.

Please check the file and remove any potentially identifying information “e.g. deviceid, subscriberid, simid, devicephonenumber, etc.

18. Supplement 2: Kinshasa IRB: Please confirm that the appropriate usage rights apply to the use of the maps on page 3 and 4 of the protocol. PLOS applies the Creative Commons Attribution (CC BY) license to all the works we publish, and the use of figures must be compatible with this license. Please see our guidelines for map images: https://journals.plos.org/plosmedicine/s/figures#loc-maps

Please note relevant permissions and license/copyright information in the legends.

[LINK]

---

## [Editor Report · Decision Letter 4]

29 Nov 2021

Dear Dr Ramaiya, 

On behalf of my colleagues and the Academic Editor, Kathryn Yount, I am pleased to inform you that we have agreed to publish your manuscript "Assessing the relationship between agency and peer-violence among adolescents aged 10-14 years in Kinshasa, Democratic Republic of Congo and Blantyre, Malawi: A cross-sectional study" (PMEDICINE-D-21-00483R4) in PLOS Medicine’s Special Issue: Global Child Health: From Birth to Adolescence and Beyond.

Please also address the following editorial requests:

1. Abstract: Lines 57-58: Please report exact p values, unless p<0.001, instead of reporting “p<0.01” or “p<0.05” in these sentences.

2. Methods: Line 269: It seems that “graffiti” should not be capitalized in this sentence.

3. Reference 11: Please abbreviate the journal title as: Int J Adolesc Med Health.

4. Reference 33: Please abbreviate the journal title as: Lancet Public Health.

5. Supplement 3: Blantyre IRB: Please include a clean version of the document, without comments/tracked changes.

6. S5 Table and S6 Table: In the legend, please note the meaning behind the # symbol used in the table (e.g. “#Malawi tertile 2”).

PRESS

Sincerely, 

Caitlin Moyer, Ph.D. 

Associate Editor

PLOS Medicine